# Effects of Xanthine Oxidase Inhibition by Febuxostat on Lipid Profiles of Patients with Hyperuricemia: Insights from Randomized PRIZE Study

**DOI:** 10.3390/nu16142324

**Published:** 2024-07-19

**Authors:** Yuichi Saito, Atsushi Tanaka, Hisako Yoshida, Hitoshi Nakashima, Noriko Ban, Munehide Matsuhisa, Yoshio Kobayashi, Koichi Node

**Affiliations:** 1Department of Cardiovascular Medicine, Chiba University Graduate School of Medicine, Chiba 260-0856, Japan; saitoyuichi1984@gmail.com (Y.S.); yuiryosuke@msn.com (Y.K.); 2Department of Cardiovascular Medicine, Saga University, Saga 849-8501, Japan; node@cc.saga-u.ac.jp; 3Department of Medical Statistics, Osaka Metropolitan University Graduate School of Medicine, Osaka 545-8585, Japan; hisako.yoshida@omu.ac.jp; 4Department of Cardiovascular Medicine, Imamura General Hospital, Kagoshima 890-0064, Japan; hnakashi35@gmail.com; 5Department of Internal Medicine, Chiba Aoba Municipal Hospital, Chiba 260-0852, Japan; nontack98081@muc.biglobe.ne.jp; 6Diabetes Therapeutics and Research Center, Institute of Advanced Medical Sciences, Tokushima University, Tokushima 770-8503, Japan; matuhisa@tokushima-u.ac.jp

**Keywords:** cholesterol, hyperuricemia, febuxostat, dyslipidemia

## Abstract

Although patients with hyperuricemia and gout often have dyslipidemia, the effects of febuxostat, a xanthine oxidase inhibitor, on their lipid profiles are unclear. Thus, we performed a sub-analysis of the randomized PRIZE study in which the effects of febuxostat on carotid atherosclerosis were investigated in patients with hyperuricemia. The participants were randomized to the febuxostat or control group. The primary endpoint of this sub-analysis was changes in the patients’ non-high-density lipoprotein cholesterol (HDL-C) levels from baseline to 6-month follow-up. Correlations between the changes in lipid profiles and cardiometabolic parameters were also evaluated. In total, 456 patients were included. From baseline to 6 months, non-HDL-C levels were significantly reduced in the febuxostat group (−5.9 mg/dL, 95% confidence interval [CI]: −9.1 to −2.8 mg/dL, *p* < 0.001), but not in the control group (−1.3 mg/dL, 95% CI: −4.4 to 1.8, *p* = 0.348). The reduction in non-HDL-C levels was more pronounced in women and correlated with changes in serum uric acid and estimated glomerular filtration rate levels only in the febuxostat group. In patients with hyperuricemia, febuxostat treatment was associated with reduced non-HDL-C levels from baseline to the 6-month follow-up compared to the control treatment, suggesting that the lipid-lowering effect of febuxostat should be considered when targeting dyslipidemia.

## 1. Introduction

Hyperuricemia is associated with the development of cardiovascular and renal diseases, such as hypertension, atrial fibrillation, heart failure, coronary artery disease, and chronic kidney disease [1]. From an epidemiological perspective, patients with elevated levels of serum uric acid (SUA) are likely to have dyslipidemia and other metabolic disorders [2], leading to atherosclerotic cardiovascular diseases. Guidelines recommend that patients with gout should be screened at least annually for cardiovascular risk factors and comorbid conditions, such as hypertension, diabetes, dyslipidemia, smoking, obesity, and renal disease, and should be managed appropriately [3]. A longitudinal cohort study in Japan showed that among healthy individuals, higher SUA levels at baseline were related to an increase in low-density lipoprotein cholesterol (LDL-C) levels at the 5-year follow-up [4]. A meta-analysis also demonstrated that higher SUA levels were progressively associated with a higher prevalence of dyslipidemia, including high triglyceride, low high-density lipoprotein cholesterol (HDL-C), and high total cholesterol levels [5]. Although it remains unclear whether there is a direct association between hyperuricemia and dyslipidemic disorders, therapeutic strategies against dyslipidemia may be important for reducing atherosclerotic cardiovascular events in patients with elevated SUA levels and gout [6]. However, it is uncertain whether uric acid-lowering medications can improve lipid profiles and reduce the risk of dyslipidemia. A meta-analysis reported that the uric acid-lowering effect of allopurinol, a xanthine oxidase (XO) inhibitor, does not translate into an improvement in serum lipid levels [7], while a single-center study showed that patients treated with febuxostat achieved better lipid profiles after 90 days than those treated with allopurinol [8]. In this study, we aimed to assess the effects of febuxostat, another XO inhibitor, on the lipid profiles of patients with hyperuricemia using data from a randomized controlled trial (RCT).

## 2. Materials and Methods

### 2.1. Study Design

This was a post hoc analysis of the PRIZE study, a prospective, open-label, blinded-endpoint trial (University Hospital Medical Information Network Clinical Trial Registry, UMIN000012911 and UMIN000041322). The study’s protocol and design are available in previous reports [9]. Briefly, patients who had asymptomatic hyperuricemia (SUA levels >7.0 mg/dL) and a maximum intima–media thickness (IMT) of the common carotid artery (CCA) of ≥1.1 mm at screening were eligible for the PRIZE study. Patients were excluded if they received SUA-lowering medications within 8 weeks before the eligibility assessment, and if they had gouty tophus or symptoms of gouty arthritis within 1 year prior to the assessment. Participants with hyperuricemia and significant carotid plaques in the PRIZE study were randomly assigned to the febuxostat group or the control (no intervention) group, and they also implemented appropriate lifestyle modifications for hyperuricemia. Randomization was stratified according to age, sex, presence of diabetes, SUA levels, and maximum CCA-IMT. In the febuxostat group, patients were treated with an initial daily dose of 10 mg, followed by an increased dose of 20 mg at the 1-month timepoint and 40 mg at the 2-month timepoint, if tolerated. Although a daily dose of 40 mg was the targeted maintenance dose, a daily dose of 60 mg was allowed at 3 months or later. In the control group, the intervention solely involved appropriate lifestyle modifications. Patients with missing data regarding non-HDL-C levels at baseline were also excluded (Figure 1).

The PRIZE study was conducted in compliance with the Declaration of Helsinki, and this sub-analysis of the PRIZE study was approved by the Ethics Committee Saga University Hospital (2020-05-R01). All the participants of the PRIZE study provided written informed consent.

### 2.2. Measurements and Outcomes

Blood samples were obtained at baseline and after 6, 12, and 24 months. The levels of non-HDL-C, total cholesterol, HDL-C, LDL-C, and triglycerides were measured. SUA levels, body mass index (BMI), systolic blood pressure (SBP), and estimated glomerular filtration rate (eGFR) were also measured and recorded at each timepoint. The primary endpoint of the present sub-analysis was the non-HDL-C level [10], and the primary interest was determining the impact of febuxostat treatment on lipid profiles. Because non-HDL-C and SUA levels changed significantly from baseline to 6 months and did not differ significantly thereafter, we mainly focused on the changes up to the 6-month follow-up. Correlations between changes in non-HDL-C and cardiometabolic variables (i.e., SUA levels, BMI, SBP, and eGFR) from baseline to 6 months were also evaluated.

### 2.3. Statistical Analysis

Statistical analysis was conducted by an independent biostatistician using R statistical software, version 4.0.2 (R Foundation for Statistical Computing, Vienna, Austria). All continuous variables were shown as medians [interquartile ranges], and categorical variables as frequencies (percentages). The standard mean difference between the febuxostat and control groups was calculated. We used linear mixed-effects models, including treatment group and time interaction terms, to estimate group differences at each timepoint, and compared the trajectories of lipid profiles (non-HDL-C, total cholesterol, HDL-C, LDL-C, and triglycerides), SUA levels, BMI, SBP, and eGFR between adjacent timepoints. Effect modification, which examined whether the effect of the treatment group on the non-HDL-C trajectory was influenced by the levels of a covariate, was evaluated using interaction analysis. This involved the introduction of a cross-product term between the treatment group and a binary variable representing a subgroup of patients in the mixed-effects model. Forest plots were used to visually represent the effects of modifications. Correlations between changes in non-HDL-C and cardiometabolic variables from baseline to the 6-month follow-up were also assessed using mixed-effects models that incorporated treatment group and time interaction terms. Statistical significance was set at *p* < 0.05.

## 3. Results

From May 2014 to June 2016, 514 patients were registered in the PRIZE study and randomized to the febuxostat or control group, of whom 456 (229 and 227 in the febuxostat and control groups, respectively) were included in the present study (Figure 1). Their baseline characteristics are listed in Table 1. Overall, the median age was 71 years; men accounted for more than 80% of the study population; and cardiovascular risk factors, particularly hypertension, were prevalent (Table 1). The median non-HDL-C level was 128.0 mg/dL, with statins in 48.5% of patients, eicosapentaenoic acid in 3.5%, ezetimibe in 3.3%, and fibrates in 1.8% at baseline. The two groups were well balanced (Table 1). Details of the lipid-lowering drugs used at each timepoint are listed in Table 2, which shows that changes in the two groups and between-group differences in medication use were small.

From baseline to 6 months, non-HDL-C, total cholesterol, LDL-C, and SUA levels were significantly reduced in the febuxostat group, but not in the control group (Figure 2). No significant changes were found in other cardiometabolic variables, such as HDL-C levels, triglycerides, SUA levels, BMI, SBP, and eGFR, at the 6-month follow-up (Figure 2). Between-group differences in non-HDL-C, total cholesterol, triglyceride, and SUA levels from baseline to the 6-month follow-up were significant (Table 3). Subgroup analysis revealed that the non-HDL-C-lowering effect was likely to be greater in women treated with febuxostat (Figure 3). Changes in non-HDL-C levels from baseline to 6 months were significantly correlated with changes in SUA levels and eGFR in the febuxostat group, but not in the control group (Figure 4). The between-group interactions were not significant for the correlation of changes in non-HDL-C levels with changes in SUA levels, BMI, SBP, and eGFR (Figure 4). Similarly, changes in non-HDL-C levels from baseline to the 6-month follow-up were significantly correlated with changes in SUA levels and eGFR in the febuxostat group, but not in the control group (Appendix A).

## 4. Discussion

In this sub-analysis of the PRIZE study, the febuxostat treatment was associated with reduced non-HDL-C levels from baseline to the 6-month follow-up compared with the control treatment in patients with elevated SUA levels. The reduction in non-HDL-C levels was more evident in women and correlated with changes in SUA levels and eGFR in the febuxostat group. The lipid-lowering effect of febuxostat may translate into better care of patients with elevated SUA levels.

### 4.1. Uric Acid Lowering and Lipid Profiles

Although causality has not yet been established, patients with elevated levels of SUA are at a high risk of cardiovascular events. This association may be partly due to the direct effects of uric acid, including inflammation, oxidative stress, and endothelial dysfunction, while concomitant cardiovascular risk factors in patients with elevated SUA levels and gout presumably play significant roles in the development of cardiovascular and renal diseases [1]. Thus, therapeutic strategies against dyslipidemia (e.g., statins) may be important in improving clinical outcomes in this patient population [6]; however, the effects of uric acid-lowering medications on lipid profiles remain unclear, particularly in the case of newer XO inhibitors. A systematic meta-analysis of seven prospective studies showed that allopurinol (100–300 mg daily) reduced SUA levels by approximately 2 mg/dL, while no effects on lipid profiles, including total cholesterol, HDL-C, LDL-C, and triglyceride levels, were found at the 24-month follow-up [7]. Interestingly, several studies have indicated the beneficial effects of allopurinol on oxidized LDL levels [11,12]. A randomized, double-blind, placebo-controlled crossover study (*n* = 80) showed that in patients with stable obstructive coronary disease, a high dose of allopurinol (300–600 mg daily) was associated with improved systemic endothelial function—as evaluated by flow-mediated dilation—and a relative reduction in oxidized LDL after 8 weeks of treatment [12], suggesting the potential of using XO inhibition to ameliorate dyslipidemic conditions. However, data on new XO inhibitors, including febuxostat and topiroxostat, are limited. A single-center, non-randomized study in China (*n* = 60) indicated that patients treated with febuxostat (80 mg daily) had lower total cholesterol (−16.8 ± 14.8 vs. 2.9 ± 4.1 mg/dL, *p* < 0.001) and LDL-C (−15.1 ± 2.7 vs. 2.5 ± 7.0 mg/dL, *p* < 0.001) levels at the 90-day follow-up than those treated with allopurinol (300 mg daily) [8]. In another retrospective, single-arm observational study in Japan (*n* = 83), the initiation of treatment with topiroxostat was associated with a reduction in total cholesterol and LDL-C levels in patients with hyperuricemia, with changes from baseline to 24 weeks of −9.1 ± 26.6 mg/dL (*p* < 0.01) and −9.1 ± 24.1 mg/dL (*p* < 0.01), respectively [13]. In a Japanese study, a reduction in total cholesterol levels significantly correlated with reduced SUA levels (*r* = 0.23, *p* < 0.05) [13], which may be in line with our results. However, the robustness of these findings is limited because of the non-randomized and retrospective study designs with small sample sizes.

### 4.2. Effects of Febuxostat on Lipid Profiles

To evaluate the pharmacological effects of febuxostat on lipid profiles, we assessed the changes in these parameters from baseline to 6 months. During this period, febuxostat treatment resulted in a relative reduction of 4.6 mg/dL in non-HDL-C levels compared with the control treatment. This reduction was mainly driven by the reduced LDL-C levels and a relative increase in triglyceride levels in the control group. Considering the established linear relation between non-HDL-C and LDL-C levels and cardiovascular outcomes, these results may be clinically relevant [14]. The present study also indicates that febuxostat’s potential effect of lowering cholesterol levels may be more evident in women. Although the underlying mechanisms are unclear, an experimental study suggested that elevated levels of serum cholesterol and the cholesterol metabolite 27-hydroxycholesterol upregulate the expression of the uric acid reabsorption transporter (URAT1), which is encoded by *SLC22A12* through estrogen receptors [15]. It is conceivable that elevated URAT1 expression increases SUA levels via reabsorption, although it remains uncertain whether the reduction in SUA levels by febuxostat treatment, in turn, reduces cholesterol levels; therefore, this warrants further investigation. In the present study, the reduced non-HDL-C levels correlated with a reduction in SUA levels and elevated eGFR at 6 months. These correlations were found in the febuxostat group, but not in the control group. The correlations were modest but suggested the presence of the URAT1 and *SLC22A12* pathways as part of the mechanisms. From the perspective of medications for dyslipidemia, atorvastatin and fenofibrate have reportedly lowered SUA levels in RCTs [16,17]. In addition, another sub-analysis of the PRIZE study demonstrated the potential effect of febuxostat in lowering the levels of malondialdehyde-modified LDL, an oxidative stress marker [18]. When considering these findings together, we believe that the association between dyslipidemia and uric acid requires further investigation. Given that, according to an expert consensus document [19], even a small increase in SUA levels can be related to a significant increase in complications in patients with hyperuricemia, uric acid-lowering therapy using XO inhibitors may be clinically relevant. In addition to that point, febuxostat treatment may be considered when treating patients with hyperuricemia and dyslipidemia, particularly women.

### 4.3. Limitations

The present study has limitations. The PRIZE study is a prospective RCT, and this sub-analysis was conducted in a post hoc manner. Owing to the open-label design of the PRIZE study, treatment decisions may have been affected by study allocation; however, the details of the medications for dyslipidemia were balanced between the febuxostat and control groups. Although the close relationships of metabolic syndrome and dyslipidemia to inflammation are well-known [20], the present study did not evaluate inflammatory biomarkers such as C-reactive protein. We used the first 6 months of the study period to evaluate febuxostat’s pharmacological potential in relation to lipid profiles, but no significant differences in these parameters were found between the two groups at 24 months (Appendix A). However, non-HDL-C levels were consistently lower in the febuxostat group than in the control group during the 24-month follow-up period, and no remarkable changes in cardiometabolic parameters were observed after 6 months. Participants in the PRIZE study had asymptomatic hyperuricemia and carotid plaques (CCA-IMT ≥1.1 mm at screening), which is a predictor of cardiovascular events [21,22]. Therefore, it is uncertain whether our results can be extrapolated to other populations.

## 5. Conclusions

In patients with hyperuricemia, the febuxostat treatment resulted in a greater reduction in non-HDL-C levels at 6 months than the control treatment in an RCT setting, particularly in women. A reduction in non-HDL-C levels was correlated with changes in SUA levels and eGFR only in the febuxostat group. The lipid-lowering effect of febuxostat should be considered when targeting dyslipidemia.

## Figures and Tables

**Figure 1 nutrients-16-02324-f001:**
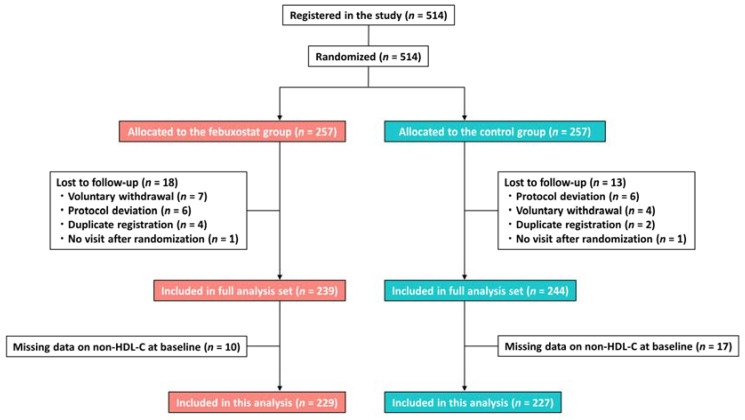
Study flow. HDL-C, high-density lipoprotein cholesterol.

**Figure 2 nutrients-16-02324-f002:**
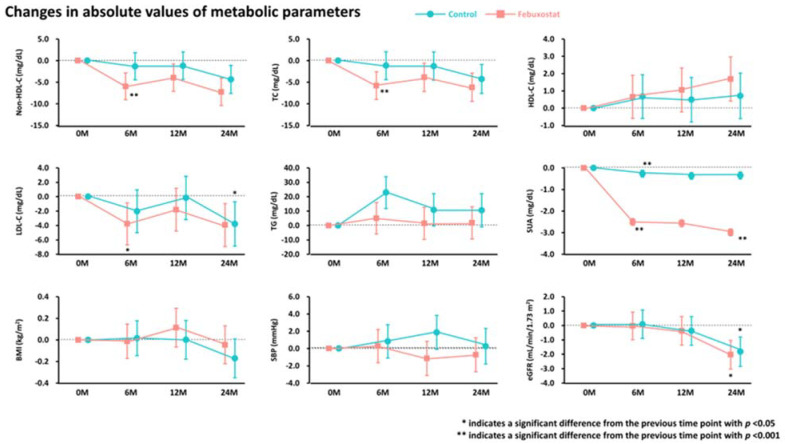
Absolute changes in metabolic parameters during the 24-month follow-up period. BMI, body mass index; eGFR, estimated glomerular filtration rate; HDL-C, high-density lipoprotein cholesterol; LDL-C, low-density lipoprotein cholesterol; SBP, systolic blood pressure; SUA, serum uric acid; TC, total cholesterol; TG, triglyceride.

**Figure 3 nutrients-16-02324-f003:**
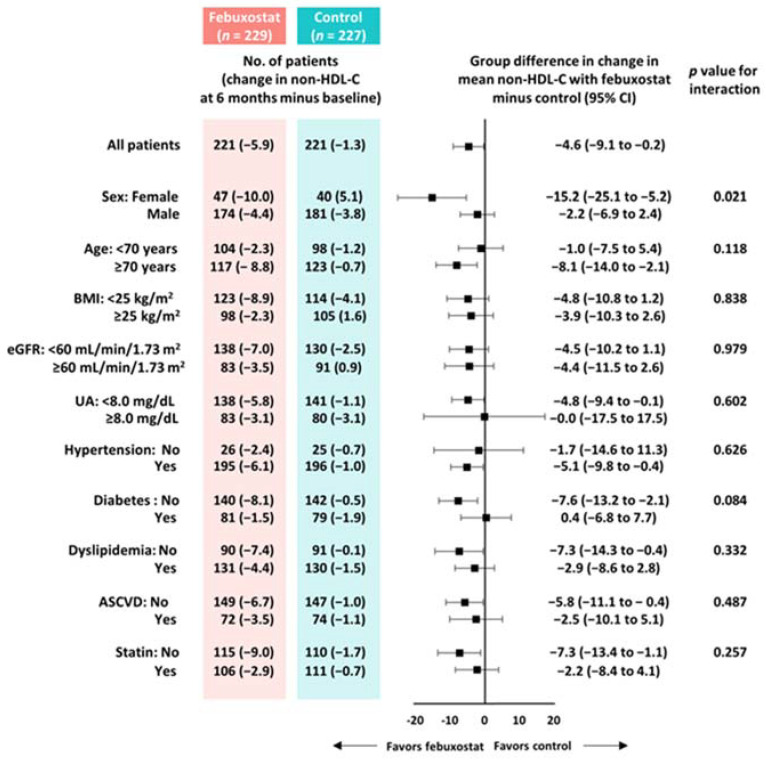
Subgroup analyses of between-group differences in changes in non-HDL-C levels from baseline to 6 months. ASCVD, atherosclerotic cardiovascular disease; BMI, body mass index; CI, confidence interval; eGFR, estimated glomerular filtration rate; HDL-C, high-density lipoprotein cholesterol; UA, uric acid.

**Figure 4 nutrients-16-02324-f004:**
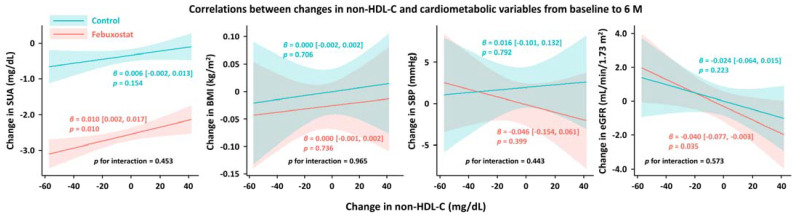
Correlations between changes in non-HDL-C levels and cardiometabolic variables from baseline to 6 months. BMI, body mass index; eGFR, estimated glomerular filtration rate; HDL-C, high-density lipoprotein cholesterol; SBP, systolic blood pressure; SUA, serum uric acid.

**Table 1 nutrients-16-02324-t001:** Baseline characteristics.

Variable	All	Febuxostat Group	Control Group	SMD
	(*n* = 456)	(*n* = 229)	(*n* = 227)	
Age (years)	71.0 [63.0, 77.0]	70.0 [63.0, 76.0]	71.0 [63.0, 78.0]	0.020
Male	367 (80.5%)	181 (79.0%)	186 (81.9%)	0.073
Body mass index (kg/m^2^)	24.6 [22.5, 27.0]	24.4 [22.3, 26.8]	24.7 [22.7, 27.2]	0.024
<18.5	15 (3.3%)	5 (2.2%)	10 (4.4%)	
18.5–24.9	231 (50.9%)	124 (54.1%)	107 (47.6%)	
25.0–29.9	167 (36.8%)	77 (33.6%)	90 (40.6%)	
30.0–34.9	32 (7.0%)	15 (6.6%)	17 (7.6%)	
≥35	9 (2.0%)	8 (3.5%)	1 (0.4%)	
Hypertension	404 (88.6%)	203 (88.6%)	201 (88.5%)	0.003
Antihypertensive drugs	415 (91.0%)	208 (90.8%)	207 (91.2%)	0.013
Diabetes	164 (36.0%)	83 (36.2%)	81 (35.7%)	0.012
Dyslipidemia	269 (59.0%)	137 (59.8%)	132 (58.1%)	0.034
Prior ASCVD	147 (32.2%)	73 (31.9%)	74 (32.6%)	0.015
eGFR (mL/min/1.73 m^2^)	55.0 [45.5, 66.7]	54.9 [44.7, 65.9]	55.4 [46.8, 66.8]	0.100
Serum uric acid (mg/dL)	7.6 [7.1, 8.2]	7.6 [7.2, 8.2]	7.6 [7.1, 8.3]	0.058
Non-HDL-C (mg/dL)	128.0 [105.5, 153.5]	128.0 [106.0, 156.0]	126.0 [105.0, 153.0]	0.053
Total cholesterol (mg/dL)	181.0 [156.0, 204.5]	182.0 [156.0, 205.0]	179.0 [155.0, 204.0]	0.034
HDL-C (mg/dL)	50.0 [42.5, 60.2]	49.0 [42.0, 60.0]	51.0 [43.0, 61.0]	0.045
LDL-C (mg/dL)	97.4 [77.6, 120.8]	97.6 [79.2, 119.6]	95.9 [74.6, 121.6]	0.034
Triglyceride (mg/dL)	125.5 [90.0, 191.5]	125.0 [93.0, 195.0]	129.0 [88.0, 182.0]	0.030

ASCVD, atherosclerotic cardiovascular disease; eGFR, estimated glomerular filtration rate; HDL-C, high-density lipoprotein cholesterol; LDL-C, low-density lipoprotein cholesterol; SMD, standard mean difference.

**Table 2 nutrients-16-02324-t002:** Lipid-lowering drugs used at each time point.

Variable	All	Febuxostat Group	Control Group	SMD
	(*n* = 456)	(*n* = 229)	(*n* = 227)	
At baseline (n)	456	229	227	
Statin	221 (48.5%)	108 (47.2%)	113 (50.2%)	0.052
Fibrate	8 (1.8%)	3 (1.3%)	5 (2.2%)	0.068
EPA	16 (3.5%)	5 (2.2%)	11 (4.8%)	0.145
Ezetimibe	16 (3.3%)	10 (4.2%)	6 (2.5%)	0.094
At 6 months (n)	448	225	223	
Statin	219 (48.9%)	109 (48.4%)	110 (49.3%)	0.018
Fibrate	9 (2.0%)	3 (1.3%)	6 (2.7%)	0.097
EPA	16 (3.6%)	5 (2.2%)	11 (4.9%)	0.146
Ezetimibe	15 (3.3%)	9 (4.0%)	6 (2.7%)	0.073
At 12 months (n)	429	217	212	
Statin	205 (47.8%)	104 (47.9%)	101 (47.6%)	0.006
Fibrate	9 (2.1%)	3 (1.4%)	6 (2.8%)	0.101
EPA	16 (3.7%)	5 (2.3%)	11 (5.2%)	0.152
Ezetimibe	15 (3.5%)	8 (3.7%)	7 (3.3%)	0.021
At 24 months (n)	407	211	196	
Statin	197 (48.4%)	101 (47.9%)	96 (49.0%)	0.022
Fibrate	8 (2.0%)	2 (0.9%)	6 (3.1%)	0.151
EPA	16 (3.9%)	4 (1.9%)	12 (6.1%)	0.217
Ezetimibe	15 (3.7%)	8 (3.8%)	7 (3.6%)	0.012

EPA, eicosapentaenoic acid; SMD, standard mean difference.

**Table 3 nutrients-16-02324-t003:** Changes from baseline in cardiometabolic variables at 6 months.

Variable	Febuxostat Group	*p*-Value *	Control Group	*p*-Value *	Group Difference	*p*-Value **
	Changes (95% CI)		Changes (95% CI)		(95% CI)	
Non-HDL-C (mg/dL)	−5.9 (−9.1 to −2.8)	<0.001	−1.3 (−4.4 to 1.8)	0.348	−4.6 (−9.1 to −0.2)	0.039
Total cholesterol (mg/dL)	−5.8 (−9.0 to −2.5)	<0.001	−1.2 (−4.4 to 2.1)	0.424	−4.6 (−9.2 to −0.0)	0.048
HDL-C (mg/dL)	0.7 (−0.6 to 1.9)	0.637	0.7 (−0.6 to 1.9)	0.711	−0.0 (−1.8 to 1.8)	0.984
LDL-C (mg/dL)	−3.8 (−6.7 to −0.8)	0.002	−2.0 (−4.5 to 0.9)	0.076	−1.7 (−5.9 to 2.4)	0.411
Triglyceride (mg/dL)	−5.1 (−5.8 to 16.0)	0.171	22.9 (11.8 to 34.0)	0.077	−17.8 (−33.1 to −2.5)	0.023
Serum uric acid (mg/dL)	−2.5 (−2.7 to −2.4)	<0.001	−0.3 (−0.4 to −0.1)	<0.001	−2.3 (−2.5 to −2.1)	<0.001
Body mass index (kg/m^2^)	−0.0 (−0.2 to −0.1)	0.738	−0.0 (−0.1 to 0.2)	0.946	−0.0 (−0.3 to 0.2)	0.801
Systolic BP (mmHg)	0.3 (−1.6 to 2.2)	0.824	0.8 (−1.1 to 2.8)	0.059	−0.5 (−3.3 to 2.2)	0.698
eGFR (mL/min/1.73 m^2^)	−0.0 (−1.0 to 0.9)	0.909	0.1 (−0.9 to 1.1)	0.997	−0.1 (−1.5 to 1.3)	0.869

* For changes; ** for groups. CI, confidence interval; BP, blood pressure; eGFR, estimated glomerular filtration rate; HDL-C, high-density lipoprotein cholesterol; LDL-C, low-density lipoprotein cholesterol; TG, triglyceride.

## Data Availability

The original contributions presented in the study are included in the article, further inquiries can be directed to the corresponding author.

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
