# Peer review of "Effects of Xanthine Oxidase Inhibition by Febuxostat on Lipid Profiles of Patients with Hyperuricemia: Insights from Randomized PRIZE Study"

_nutrients, 2024, doi:10.3390/nu16142324_

Round 1
Reviewer 1 Report
Comments and Suggestions for Authors The article studies the effect of febuxostat on lipid metabolism in hyperuricemia patients.. The abstract: please rewrite in the classic formula/structrue, with aim/results and conclussions organised and emphasized. The aim of the article is to observe the effect of xantine oxidase inhibition by febuxostat on lipid levels- a randomised study In material and methods: please add inclusion/ exclusion criteria. For each result/mechanism in the discussion, please add and shortly compare your findings and why your results are the same/different. A paragraph with treatment options/chalenges could be added and how the research could change the paradigm. The conclussions are short and closely related to the results. Please add a paragraph/discussion about inflamation-metabolic syndrome is closely related to inflamation along with MACE. The article presents a large comparison in microbiome, comparing to other articles. A brief presentation of other results/a meta-analisys could be also presented in discussion. The methods are properly used, no sugestions are to be made The conclusions are strongly related to the results, but a better soundness could be achieved with specific results (bacterial genera e.g.) The references are well chossen English grammar: no important changes are to be done, as the english language is comprehensive and properly used. Figures and tables- number, format- suitable for publication, no important duplicity in data presentation. Length of the manuscript- no inappropriate citation or recurrent data/information was foundAuthor Response
Please see the attached file.

Reviewer 2 Report
Comments and Suggestions for Authors
Dear Autora,
Congratulation on intresting study. I am glad that after initial failure in showing significant impact of Febuxostat on IMT, the subanalysis are carried on.
The matters that requires correction include disscusion regarding results. The change on non-hdl-c and TC can be result of difference in triglicerides. There was no difference between both groups on LDL cholesterol on follow-up. At 6-month follow-up febuxostat significantly decresed the the LDL-c, but simillar trend was also noticed in control with p=0.07. There was no effect of febuxostat on LDL-c at 2-years of follow-up, however there is visible trend in reduction of non-HDL, TG and TC. The overall effect could be result of impact on TG (increase at 6-month on control), leading to reaching statistical sagnificance in non-hdl-c. The impact on TG and non-HDL-c is extremely interesting and should be further investigated in high risk patients as a addition to lipid-lowering therapy in patients with elevated sUA.
Event small increase in sUA in that patients can be related to significant complications (over 5 mg/dl https://doi.org/10.5603/cj.98254). As suggested in expert consensus regarding this matter, the early treatment of sUA is important, and Febuxostat can be intresting tool.
Comments on the Quality of English LanguageMinor editing of English language required
